# Melatonin and the Brain–Heart Crosstalk in Neurocritically Ill Patients—From Molecular Action to Clinical Practice

**DOI:** 10.3390/ijms23137094

**Published:** 2022-06-25

**Authors:** Artur Bekała, Włodzimierz Płotek, Dorota Siwicka-Gieroba, Joanna Sołek-Pastuszka, Romuald Bohatyrewicz, Jowita Biernawska, Katarzyna Kotfis, Magdalena Bielacz, Andrzej Jaroszyński, Wojciech Dabrowski

**Affiliations:** 1Department of Anesthesiology in Obstetrics and Gynecology, Gynecology and Obstetrics Clinical Hospital, Poznań University of Medical Sciences, 60-535 Poznan, Poland; plotekw@ump.edu.pl; 2Department of Anesthesiology and Intensive Care, Medical University of Lublin, Jaczewskiego Street 8, 20-954 Lublin, Poland; dsiw@wp.pl (D.S.-G.); magda.bielacz@gmail.com (M.B.); 3Department of Anaesthesiology and Intensive Therapy, Pomeranian Medical University, 71-242 Szczecin, Poland; joanna.pastuszka@pum.edu.pl (J.S.-P.); romuald.bohatyrewicz@pum.edu.pl (R.B.); lisienko@wp.pl (J.B.); 4Department of Anaesthesiology, Intensive therapy and Acute Intoxications, Pomeranian Medical University in Szczecin, 70-111 Szczecin, Poland; katarzyna.kotfis@pum.edu.pl; 5Department of Nephrology, Institute of Medical Science, Jan Kochanowski University of Kielce, 25-735 Kielce, Poland; jaroszynskiaj@interia.pl

**Keywords:** brain injury, brain–heart cross talk, melatonin, inflammation, oxidative stress, critically ill, blood–brain barrier

## Abstract

Brain injury, especially traumatic brain injury (TBI), may induce severe dysfunction of extracerebral organs. Cardiac dysfunction associated with TBI is common and well known as the brain–heart crosstalk, which broadly refers to different cardiac disorders such as cardiac arrhythmias, ischemia, hemodynamic insufficiency, and sudden cardiac death, which corresponds to acute disorders of brain function. TBI-related cardiac dysfunction can both worsen the brain damage and increase the risk of death. TBI-related cardiac disorders have been mainly treated symptomatically. However, the analysis of pathomechanisms of TBI-related cardiac dysfunction has highlighted an important role of melatonin in the prevention and treatment of such disorders. Melatonin is a neurohormone released by the pineal gland. It plays a crucial role in the coordination of the circadian rhythm. Additionally, melatonin possesses strong anti-inflammatory, antioxidative, and antiapoptotic properties and can modulate sympathetic and parasympathetic activities. Melatonin has a protective effect not only on the brain, by attenuating its injury, but on extracranial organs, including the heart. The aim of this study was to analyze the molecular activity of melatonin in terms of TBI-related cardiac disorders. Our article describes the benefits resulting from using melatonin as an adjuvant in protection and treatment of brain injury-induced cardiac dysfunction.

## 1. Introduction

Brain injury is a serious and social health problem commonly associated with high disability and mortality [1]. Brain injury is not a specific disease but a multifactorial syndrome resulting in disorders of brain function. Traumatic brain injury (TBI) is the most common type of brain injury, leading to mental deterioration, behavioral disorders, disability, and/or death. However, many other factors, such as enormous stress and emotional experiences or severe general illnesses, can also result in TBI-like outcomes. Hence, brain injury has frequently been referred to as “the silent social epidemic” [2]. There are two phases of brain injury. The first phase is related to damaging factors—from completely independent factors, such as mechanical trauma, to partially modified factors, such as stress. The second phase is associated with biochemical, hormonal, and metabolic disturbances. The exacerbation of these disturbances can be controlled and modified by several drugs. These disorders may not only exacerbate primary brain injury but contribute to TBI-related damage to extracerebral organs. TBI-related extracerebral organs failure is commonly known as brain–multiorgan interaction, and it depends on the injured organ (Figure 1). According to the current recommendations, treatment of TBI is based mainly on clinical symptoms and regards all disorders in the injured brain [3,4]. Disorders in the tryptophan pathway and melatonin synthesis are underestimated problems. However, various studies have documented the crucial role of melatonin in the brain and its extremely important role in the metabolism of extracerebral organs [5,6,7,8,9,10,11,12]. Many researchers have confirmed its neuroprotective effect, as it reduces the severity of the second phase of brain injury [5,6,7,8]. Therefore, the effect of melatonin disorders on TBI-related extracerebral multiorgan injury deserves attention.

### 1.1. Melatonin

Melatonin, or 5-methoxy-N-acetyltryptamine, is a natural neurohormone widely used in medicine and commonly known as a sleep-correcting medication. Its concentration in the blood depends on the circadian rhythm. The highest secretion from the pineal gland is observed at night [13,14]. Physiologically, melatonin is mainly secreted by the pineal gland. However, the intestines, skin, retina, testes, ovaries, glial and microglial cells, and lymphocytes can also synthetize and release melatonin [14,15,16,17]. Its production and metabolism depend on race, age, and gender, and the highest melatonin concentration was found in epidermis of a young African American population [18]. However, the pineal gland is the most important place for melatonin synthesis.

Melatonin is synthesized from tryptophan, which is hydroxylated to 5-hydroxytryptophan. This process is regulated by the light/dark cycle, and blue light (i.e., light waves between 420 and 480 nm) plays a crucial role via suppression of arylalkyl-amine N-acetyltransferase (AANAT) (Figure 2). Therefore, melatonin synthesis is better at night and with dark conditions, whereas light inhibits its synthesis and secretion [14,15]. Next, 5-hydroxytryptophan is decarbonized to serotonin, which is subsequently transferred to melatonin (Figure 2). Serotonin is a precursor to melatonin via sequential acetylation to N-acetyloserotonin (NAS) by AANAT or arylamine N-acetyltransferase (NAT) following methylation by hydroxyindole-O-methyltransferase (HIOMT) [19,20,21,22]. NAT and AANAT are enzymes that exhibit a circadian rhythm, similarly to melatonin [23,24]. Two functional isoenzymes, NAT1 and NAT2, are present in humans, and both possess strong capabilities to modify cellular functions, including anticancer activity [24]. An experimental study showed the presence of NAT and the last enzyme of serotonin metabolism, HIOMT, in rat cortical astrocytes, which confirmed that rat cortical astrocytes synthetized melatonin via the traditional synthetic pathway [25].

Serotonin and NAS are produced endogenously in the pineal gland as well as in peripheral organs such as the liver and the epidermal, dermal, and adnexal compartments of human skin. Both were found in human serum, and serotonin levels were 2.6 times higher than those of NAS in the epidermis and 121 times higher in the serum [26]. It was documented that serum serotonin concentrations did not depend on age or gender, whereas NAS concentrations were slightly higher in females than in males [27]. Interestingly, clinical observations showed dramatically lower serum NAS concentration in children with attention disorders, and changes in its concentrations corresponded to the sleep cycle and serum melatonin, which might suggest that serum NAS results from its release from neural tissue [27]. Production of serotonin in the epidermal cells can be intensified by ultraviolet light (UV), and NAS in human serum may be a substrate for enzymatic transformation to melatonin, especially in the liver and skin [19,22,28].

Melatonin possesses broad-spectrum activity, as it synchronizes the circadian rhythm, regulates energy metabolism and the ageing process, attenuates blood–brain barrier (BBB) impairment, and modulates the neuroinflammatory response and oxidative stress in the brain and peripheral tissues [14,15,26,29,30,31,32,33,34]. Melatonin also affects the peripheral nervous system. The administration of melatonin attenuated the reflex of cutaneous vasoconstriction in response to cold stress [35]. Additionally, it modulated the neural control of the sympathetic reflex induced by postural changes and attenuated increased skin sympathetic nerve activity following mental stress [36,37].

Melatonin plays a crucial role in the circadian rhythm. It possesses strong antioxidant, anti-inflammatory, and antiapoptotic properties, which work at the cell respiration level by promoting mitochondrial oxidative phosphorylation [13,14]. Interestingly, NAS, a precursor of melatonin, demonstrated a stronger antioxidant effect than melatonin, especially against iron, copper, H_2_O_2_, and lipid peroxidation [38]. Melatonin activates two specific receptors, MT1 and MT2, which are found not only in the brain but in the extracerebral organs [5,10,14,34,39,40,41,42,43,44]. The M1 receptor was found in humans nearly 20 years ago by Reppert and colleagues [45] and is expressed in several regions of the brain, such as the hypothalamus, hippocampus, cerebellum, ventral tegmental area, and substantia nigra, as well as in extracerebral cells such as the peripheral blood vessels, heart and aorta, immune system, breast, and pancreas [45,46,47,48]. One year later, the MT2 receptor was found in the brain, retina, and pituitary gland [44,47,48]. Both belong to the family of G protein-coupled transmembrane receptors, which contain the 7α helical transmembrane domains [49]. A third membrane-bound melatonin receptor, called MT3, was found in the liver, kidney, heart, and brain of hamsters [44,50].

Melatonin is metabolized in various concurrent catabolic pathways, including enzymatic ones (the indolic and kynuric pathways), and direct, nonenzymatic conversions (Figure 2) [20,21,51]. The half-life of plasma melatonin is approximately 40 min [52]. The degradation of melatonin occurs at the site of its synthesis and in the liver, where the circulating melatonin is metabolized by hepatic cytochrome P-450 enzymes, especially by CYP1A1, CYP1A2, CYP2C19, and CYP2C9 isoforms [51,53,54]. Most of the circulating melatonin is metabolized in the liver to 6-hydroxymelatonin and subsequently to 6-hydroxymelatonin sulphate. CYP1A2 plays a major role in this process; however, CYP1A1 and the extrahepatic isoform of P-450, CYP1B1, also contribute to the formation 6-hydroxymelatonin [54]. An experimental study documented that 40% of melatonin was also hydroxylated by CYP1B1 in the cerebral cortex [55]. Another isoenzyme from the P-450 family, CYP2C19, can demethylate melatonin to its precursor NAS [55]. In the kynuric pathway, melatonin is metabolized mostly to N1-acetyl-N2-formyl-5-methoxykynuramine (AFMK) and subsequently to N1-acetyl-5-methoxykynuramine (AMK). Interestingly, both AFMK and AMK also possess strong antioxidant properties [56,57,58]. Melatonin is also metabolized in a nonenzymatic pathway in direct reaction with reactive oxygen and nitrogen species [31,51]. It possesses high affinity to free radicals, and its interaction with hydroxyl radical (OH^−^) is its best recognized reaction. Melatonin hydroxylation by two hydroxy radical species leads to the formation of 2-hydroxymelatonin and 4-hydroxymelatonin, which are converted to AFMK [51,59].

Melatonin metabolism can be intensified by ultraviolet (UV) radiation. Exposure to UV light with the wavelength of 280–320 nm increased melatonin metabolism and production of 2-hydroxymelatonin, 4-hydroxymelatonin, and AFMK, whereas UV irradiation did not intensify the production of 6-hydroxymelatonin, which is the main metabolite of circulating melatonin [22]. Interestingly, the reaction between melatonin and two OH^−^ in the presence of Fe+2 and UV led to the formation of an unusual molecule—cyclic 3-hydroxymelatonin, which was detected in human urine [60]. Hence, cyclic 3-hydroxymelatonin has been proposed as an ideal biomarker for monitoring in vivo changes in OH¯ generation. Additionally, the chemical structure of cyclic 3-hydroxymelatonin is like that of acetylcholinesterase inhibitors, medications improving brain recovery after TBI [61]. Many studies have documented the antioxidative and anti-inflammatory effects of melatonin and its metabolites. However, only a few of them have described the effect of melatonin on heart function, and none of them have analyzed the effect of melatonin on the brain–heart interaction in patients treated for TBI. This study discusses the possible effect of melatonin on TBI-related cardiac disorders, known as the brain–heart interaction.

### 1.2. Brain Injury and Melatonin

The administration of melatonin can be beneficial in various cerebral diseases, including ischemia, trauma, subarachnoid hemorrhage, and COVID-19 [5,7,8,62,63]. The circadian blood melatonin concentration decreases following TBI, and this disorder depends on the severity of the TBI [64,65]. The decrease observed in nonsurvivors is significantly less than that in survivors and corresponds to the six-month Glasgow Outcome Score—Extended [65,66]. A similar effect has been observed in patients treated for subarachnoid hemorrhage (SAH), and the disorders of the circadian rhythm in melatonin synthesis depend on the patient’s age and the region of bleeding. Patients with aneurysms located in posterior circulation had significantly lower blood melatonin concentrations than those with aneurysms in anterior circulation [67]. On admission to hospital, the highest blood melatonin concentration was noted in patients with poor clinical condition, those who had anterior communicating artery aneurysms, and those without external ventricular drains [67]. These finding suggested an important role of the hypothalamus in melatonin synthesis and release. Interestingly, an experimental study documented life-threatening changes in ECG with ST depression and myocardial necrosis after anterior hypothalamic stimulation [68]. The hypothalamic paraventricular nucleus controls the hypothalamic–pituitary–adrenal axis and stimulates the pituitary gland to release adrenocorticotropic hormone during stress, which increases the risk of electrocardiographic abnormalities [69]. Melatonin inhibits adrenocorticotropic-stimulated stress hormone release [70]. It also reduces the degree of TBI-related inflammatory response, neuronal apoptosis, and oxidative stress [8]. A study on Wistar rats with TBI showed that melatonin administrated at a dose of 10 mg/kg body weight attenuated neuronal apoptosis and corrected mitochondrial perturbation by inhibiting mitochondrial fission [71]. Another experimental study showed that melatonin caused selective removal of damaged mitochondria by activating mitophagy, which was an important mechanism for reducing the severity of neuroinflammation after TBI [72]. As TBI activates the neuroinflammatory pathway with uncontrolled proinflammatory cytokine production, which leads to cerebral edema and BBB disruption, the balance between anti-inflammatory and proinflammatory responses is essential for treatment. Treatment with melatonin attenuated the BBB permeability and degradation of tight-junction and adherent-junction proteins by inhibition of the TLR4/MF-κB signaling pathway and matrix metalloproteinase-9 [32,34]. An experimental study showed that melatonin also attenuated stress-related neuronal damage, neuroinflammation, and neurodegeneration in the brain and skin through inhibition of the hypothalamic–pituitary–adrenal gland (HPA) axis [73]. Furthermore, prophylactic melatonin administration inhibited endoplasmic reticulum stress and autophagy via suppression of the Jun-N-terminal kinase pathway, which is one of the major triggers for the mitogen-activated protein kinase (MAPK) inducing apoptosis [74]. Notably, endoplasmic reticulum stress and autophagy are among the main molecular pathomechanisms of cerebral ischemia/reperfusion injury [74].

An injury of the hypothalamic region may also disturb the circadian rhythm, because the suprachiasmatic nucleus is the center responsible for the control and synchronization of daily physiological and behavioral rhythms. The suprachiasmatic nucleus generates an impulse signal, which is a trigger for melatonin synthesis and release from the pineal glands [12,13,14,15]. Melatonin synthesis disorders resulting from a decreased supply of tryptophan or an uncontrolled increase in its metabolism in the kynurenic pathway may lead to insomnia, depression, or other neuropsychiatric diseases. It has been shown that brain injury followed by neuroinflammation or severe general inflammation disturbs the balance of tryptophan metabolism, which increases the synthesis of kynurenines and kynurenic, picolinic, and quinolinic acid [75,76,77]. An experimental study proved that TBI upregulated the balance in tryptophan metabolism with an increased kynurenine level at an earlier phase after injury and decreased serotonin and melatonin synthesis [78]. A decreased level of melatonin induces sleep deprivation and reduces anti-inflammatory and antioxidative abilities, not only in the brain but in extracerebral organs including the heart. Sleep deprivation per se poses a high risk of cardiac dysfunction with life-threatening cardiac arrhythmias [79,80]. Notably, cardiac dysfunction and various electrocardiographic abnormalities are often observed in the first week after the TBI onset and may correspond to a post-traumatic decrease in the blood melatonin concentration [81]. Although there are no data documenting the relationships between melatonin deprivation and TBI-related cardiac dysfunction, the maintenance of adequate blood melatonin concentration seems to be an important treatment for reducing TBI-induced disorders in the brain as well as related cardiac dysfunction.

### 1.3. The Brain–Heart Interaction

Complications and mortality after TBI can result from both the brain injury itself and TBI-related multiorgan damage, including cardiac dysfunction. This interaction is commonly known as the brain–heart axis or the brain–heart crosstalk and was first described by Walter Cannon in 1942 [82]. He suggested that cardiac disorders could result from extreme emotions and brain dysfunction, which are associated with persistent excessive activity of the sympathoadrenal system and uncontrolled catecholamine release. He based his hypothesis on numerous case reports and experimental studies describing cardiac disorders following upregulation of the autonomic system. Indeed, several researchers studied such relationships approximately half a century before Cannon’s work. In 1913, Alfred Goodman Levy observed that electrical stimulation of the sympathetic cardiac nerve or excessive emotional stress could trigger cardiac arrhythmia and ventricular fibrillation [83]. Since his study, the upregulation of the autonomic system and sympathetic hyperactivity have been suggested as the main pathomechanisms of cardiac dysfunction following brain injury [84,85,86]. The brain can affect the cardiovascular system through the autonomic nervous system and afferent and efferent activation of the sympathetic and parasympathetic nerves. The rostral ventrolateral medulla plays a crucial role in the regulation of cardiac function and the cardiovascular system. Sympathetic impulses are generated in the cingulate and orbitofrontal cortices, amygdala, parietal somatosensory cortex, and nucleus ambiguous and are transmitted to the heart through the rostral ventrolateral medulla and the intermediolateral cell column in the upper thoracic spinal cord. The parasympathetic innervation of the heart is associated with the vagus nerve, which is formed by similar structures of the brain and periaqueductal grey, the parabrachial nucleus and the dorsal motor nucleus of the vagus nerve [87]. An experimental study showed that hypothalamic stimulation was associated with increased sympathetic nerve traffic to the heart and a lowered ventricular fibrillation threshold [88]. Prolonged stimulation of the left insula cortex caused bradycardia with ST segment depression and increased plasma norepinephrine concentration, whereas stimulation of the right insula led to tachycardia [89]. This relationship was confirmed in clinical observation, which showed a significantly higher incidence of life-threatening cardiac arrhythmias and elevated troponin levels in patients with ischemic stroke of the right insula and right inferior parietal lobe [90]. An analysis of 730 patients with acute ischemic stroke showed two regions (the right insula, especially the posterior, superior, and medial areas, and the right parietal lobule) for which damage was associated with stroke-related myocardial injury [91]. This fact suggests that ischemia of the right posterior insula probably disinhibits other centers, increasing cardiac sympathetic activity and leading to myocardial injury. Interestingly, all these regions are rich in melatonin receptors [39,43].

HPA axis dysfunction is considered an important pathomechanism of TBI-related cardiac dysfunction. The HPA axis is the main regulator of body hormones, and its activity corresponds to emotion, degree of stress, physical activity, and metabolism. The activity of the HPA axis is controlled by the hypothalamic paraventricular nucleus, which is considered the main control center, secreting corticotropin-releasing factor and vasopressin. Anatomically, the hypothalamic paraventricular nucleus closely adjoins to the rostral ventrolateral medulla, which integrates cardiac afferents and coordinates the activity of the baroreceptors and autonomic nerves that regulate cardiac function. Reduced activity of the paraventricular nucleus improved cardiac function and reduced the area of myocardium apoptosis after myocardial infarct [92]. Interestingly, an experimental study showed a drastic inhibition of melatonin synthesis and release from the pineal gland after prolonged stimulation of the hypothalamic paraventricular nucleus [93]. On the other hand, an increase in melatonin levels in the hypothalamic paraventricular nucleus ameliorated myocardial injury after acute ischemia [94]. This fact may have strong implications for the brain–heart interaction.

TBI-related myocardial dysfunction may also be caused by catecholamine excess resulting from acute injury or mental stress. Additionally, brain injury increases sympathetic tone, which causes additional catecholamine release from the synapses and adrenal glands [95]. An experimental study showed rapid increases in norepinephrine and epinephrine levels as early as 5 min after the onset of subarachnoid hemorrhage [96]. The increase in these catecholamines was associated with the concentrations of plasma troponin T and myocardial creatine kinase (CK-MB). The pathomechanisms of hypercatecholamine-induced myocardial injury are multifactorial and include epicardial microartery spasm, increases in myocardial contractility with associated transient obstruction of the left ventricle outflow and intricate myocyte calcium (Ca^+2^) overload, increased free radical production, apoptosis, and myocyte death [97]. However, an experimental study suggested that myocardial damage induced by severe brain injury and brain death was rather related to endogenous catecholamine release and excessive Ca^+2^ uptake by stimulated myocytes [97]. Catecholamine increases the strength and rate of heart contraction by stimulating β-receptors. Norepinephrine exhibits higher affinity to β1 than to β2 receptors. Additionally, β receptors are distributed differently in the heart and the apex, the latter of which is characterized by a greater number of β-adrenergic receptors and lower sympathetic innervation [98]. The activation of β receptors couples G proteins and activates adenylyl cyclase to produce cytosolic cAMP, which binds to protein kinase A. This connection triggers the phosphorylation of sarcolemmal L-type calcium channels and phospholamban, increasing the uptake of Ca^+2^ by mitochondria. Notably, extensive mitochondrial calcium overload induces oxidative stress, osmotic stealing, and apoptosis. Hence, prolonged overexcitation of β receptors may result in cardiac damage.

## 2. Possible Molecular Mechanisms of Melatonin Protection

### 2.1. Cerebral Effect of Melatonin on TBI-Related Cardiac Dysfunction

Stress promotes melatonin biosynthesis and release to the blood, and low-intensity stress is a beneficial factor for the human organism [56,99]. Besides the beneficial effect of low-intensity stress, prolonged or high-intensity exercise causes several biochemical and hormonal disturbances, with massive inflammation and reactive oxygen species (ROS) production [100]. It is well recognized that TBI induces one of the most excessive inflammatory and oxidative responses. Melatonin possesses neuroprotective properties in its broad sense, as it reduces TBI-induced neuroinflammation, production of free radicals, and apoptosis. Its high free radical-scavenging capability protects the mitochondria from DNA damage induced by the accumulation of ROS [101]. Melatonin reduces ROS-induced depolarization of the mitochondrial membrane potential, which effectively prevents mitochondrial swelling, opens mitochondrial permeability transition pores, and releases cytochrome C. This effect is like the pharmacological suppression of Ca^+2^ regulatory mechanisms [101]. During the second phase of TBI, neurons and astrocytes experience a rapid increase in Ca^+2^, which is a trigger for apoptotic and necrotic cell death. Melatonin plays a physiologically relevant role in the regulation of transient receptor potential M2 (TRPM2), which is a unique, calcium-permeable, nonselective cation channel responsible for Ca^+2^ entry to neurons and astrocytes [102]. It also reverses acidosis-induced neuronal injuries and death though reductions in tau phosphorylation and kinase/phosphatase activity, intracellular free radical concentration, and acidosis-induced synaptic abnormalities [101,103]. The antioxidative effect of melatonin also results from the stimulation of several antioxidative enzymes, such as superoxide dismutase and glutathione peroxidase, and the downregulation of pro-oxidative enzymes [8,12,33,103,104]. These effects seem to be crucial for the regulation of TBI-induced energy imbalance and reduction in the degree of neuroinflammation, neurodegeneration, and neuronal apoptosis. Moreover, melatonin treatment can reverse the damaging effect of ROS and neuroinflammation in injured ipsilateral cortices and hippocampi because it can improve synaptic function by regulation of synaptic protein levels [8]. The accumulation of ROS in mitochondria can be modulated by reactive nitric oxide anions (NO^−^) generated by the activity of mitochondrial nitric oxide synthase [95]. Low NO^−^ levels inhibit the production of ROS, whereas high NO^−^ levels inhibit H_2_O_2_ production. This process is controlled by NO^−^-induced modulation of oxygen consumption via cytochrome C oxidase activity [105,106]. Melatonin has the capacity to scavenge several ROS, such as the hydroxyl radical, H_2_O_2_, O_2_^−^, NO, NOO¯, the hypochlorite radical, singlet oxygen, and others [107]. However, the main melatonin metabolites, AFMK and AMK, also possess antioxidative properties and probably present higher antioxidant activity than melatonin per se [108,109,110]. An in vitro study showed that a precursor of AMK, AFMK, protects DNA from oxidative damage and ameliorates hippocampal neuronal injury caused by glutamate excitotoxicity, H_2_O_2_, and amyloid-β peptide [108]. Interestingly, the antioxidant properties of AFMK are also associated with donation of electrons, reducing and neutralizing free radicals’ potential [107]. AMK, which is a product of AFMK conversion, can inhibit neuronal nitric oxide synthase (nNOS) through an interaction with Ca^+2^-calmodulin, which plays a crucial role in various neural functions, including cognition [111,112]. Hence, experimentally administrated AMK improved memory independently from animals’ age [58]. Interestingly, many brain diseases are associated with cardiac dysfunction, and improvement in cognitive function attenuates cardiac disability [113,114].

The administration of melatonin to the paraventricular nucleus not only improves superoxide dismutase activity and corrects NO^‾^ levels but reduces the levels of interleukin 1β and norepinephrine, inhibiting sympathetic nerve activity and thus protecting the injured myocardium [94]. Interestingly, the suppression effect is dose dependent. The intravenous injection of melatonin at a dose of 1–2 mg suppressed the activity of adrenal nerves in the hypothalamic suprachiasmatic nucleus, whereas injection at a dose of 10–20 mg enhanced their activity [115]. It has been shown that melatonin synthesis strongly depends on the activity of adrenoreceptors. However, the melatonin synthesis by the pineal gland can be intensified by corticosterone only after the activation of the β-adrenergic receptors in septic rats [116]. Notably, the density of β-adrenoreceptors is three times greater than the density of α_1_-adrenoreceptors, so electrical-stimulated hormonal production can be completely abolished after β-adrenergic blockade [117]. The overexcitation of cerebral adrenoreceptors leads to neuroinflammation and induces oxidative stress and neuronal apoptosis, and even the nonselective blockade of β-adrenoreceptors results in anti-inflammatory and antiapoptotic effects [118,119]. A similar effect was observed after melatonin treatment [72]. Stress also activates the HPA axis, and the overexcitation of the hypothalamus and pituitary gland corresponds to the circadian secretion of melatonin [120]. Melatonin inhibits upregulated hypothalamic and pituitary activities, which are well known mechanisms reducing the stimulation of adrenal glands to release catecholamine [91,92,121]. Clinical observations have shown increased endogenous melatonin concentration in the cerebrospinal fluid, but not in the blood, in response to oxidative stress and inflammation [65,122]. The unchanged blood melatonin concentration, with its significantly elevated concentration in the cerebrospinal fluid after brain injury, may suggest increased melatonin release and activity to protect only the brain and not the extracerebral tissues. Although increased intracerebral melatonin concentration can modulate the autonomic activity in the brain, and thus subsequently ameliorate damage to extracerebral organs, melatonin can also directly protect extracerebral organs. Therefore, its supplementation may also reduce brain injury-related cardiac dysfunction.

### 2.2. Direct Cardiac Protection

Melatonin also exhibits strong and direct cardioprotective activity. Its daytime serum concentrations were lower in patients with hypertensive cardiomyopathy, who developed heart failure and left ventricular hypertrophy. On admission to hospital, the serum melatonin concentration can be used as an independent predictor of left ventricular remodeling in patients treated for acute myocardial infarct [123,124]. Adjuvant treatment with melatonin has been suggested as an effective adjuvant therapy in patients with myocardial infarct. Its administration decreased the infarct severity in patients undergoing percutaneous coronary angioplasty due to ST-segment elevation myocardial infarct (STEMI) [125]. The prolonged, but not single, administration of melatonin at the dose of 2–2.5 mg reduced blood pressure, especially in nondipper hypertensive patients [126,127]. Several potential mechanisms of melatonin cardioprotective activity have been described. Melatonin reduces the blood catecholamine concentration mainly by reducing the activity of the HPA axis [91,92,112,121]. It is well known that circulating catecholamine induces tachycardia and vascular tone with increased arterial contractility and increases myocardial affinity on cardiac arrhythmia. An increase in cardiac microarterioles may induce cardiac ischemia and then ROS generation. The generation of oxygen species impairs myocardial contractility and predisposes to cardiac arrhythmias. Clinical observations seem to have confirmed such mechanisms. The use of antioxidants improves reperfusion and significantly reduces reperfusion-related life-threatening cardiac arrhythmias such as atrial fibrillation-triggered premature ventricular contractions and ventricular tachycardia [128]. Hence, the cardioprotective activity of melatonin corresponds to its antioxidative properties and is associated with the activation of the membrane melatonin receptors MT1 and MT2 [125]. The MT1 receptor in the myocardium of the left ventricle was activated practically immediately after the induction of ischemia, whereas the expression of MT2 tended to increase during the four-week follow-up period. Moreover, ischemia decreased plasma melatonin concentrations, which may have triggered its increased production and greater sensitivity of the melatonin receptors in the left ventricle. This might suggest that endogenous melatonin plays an important role in protecting the heart against acute ischemia injury, whereas its deficit in the blood should be reduced by oral or intravenous administration [129,130].

Melatonin significantly improves cardiac hemodynamic function after ischemia. The oral administration of melatonin at the doses of 10 mg and 20 mg for five days before elective coronary artery bypass graft surgery (CABG) significantly improved postoperative ejection fraction (EF) one day after surgery by 6.5% and 10.7%, respectively [131]. Better hemodynamic function corresponded to lower troponin I, NO, interleukin 1β, and caspase-3 concentrations. An experimental study confirmed clinical observations, documenting a profitable effect of melatonin and its metabolite 6-hydroxymealtonin on cardiac contractility [132]. Administration of melatonin or 6-hydroxymelatonin with a dose of 2.5 µg/h 24 h before and continuously during doxorubicin treatment significantly improved doxorubicin-induced dysfunction of left ventricular diastolic and systolic function and survival time. The postmortem analysis documented a dramatic reduction in doxorubicin-induced apoptosis of cardiomyocytes. Clinical observation showed that preoperative administration of melatonin for 3 days reduced the release of other proinflammatory cytokines, such as interleukin-6 and -9, in the postoperative period [133]. Several studies have demonstrated the anti-inflammatory properties of melatonin, but the molecular pathomechanisms of this activity have not been well documented yet. Melatonin and its metabolites downregulate cyclooxygenase 2 and selectively inhibit protein 52 (p52) acetylation and NO synthase expression, which plays a crucial role in inflammation [134]. Melatonin can also regulate the intracellular Ca^+2^ level in myocytes via MT2 interaction with intracellular proteins such as calmodulin or tubulin [135]. The modulation of membrane action potential and the expression of connexin 43 (Cx43), which is the main component of gap junction channels in the ventricles, play crucial roles in oxidative stress regulation and arrhythmic triggers in the ischemic heart [136,137]. Notably, general inflammatory response and cardiac inflammation have been proposed as a pathomechanism leading to brain injury-related cardiac dysfunction [138,139]. Therefore, adjuvant treatment with melatonin should be commonly applied to prevent cardiac dysfunction after brain injury.

## 3. Conclusions

Melatonin possesses anti-inflammatory, antioxidative, and antiapoptotic properties. Treatment with melatonin reduces brain injury-related neuroinflammation and reactive oxygen species production and attenuates neuronal apoptosis. Melatonin also reduces ischemia/reperfusion cardiac damage and inflammation. Based on the available data and the molecular properties of melatonin, we can suggest using melatonin at a daily dose of 10 mg in prevention and treatment of brain injury-related cardiac dysfunction, which is commonly known as the brain–heart interaction.

## Figures and Tables

**Figure 1 ijms-23-07094-f001:**
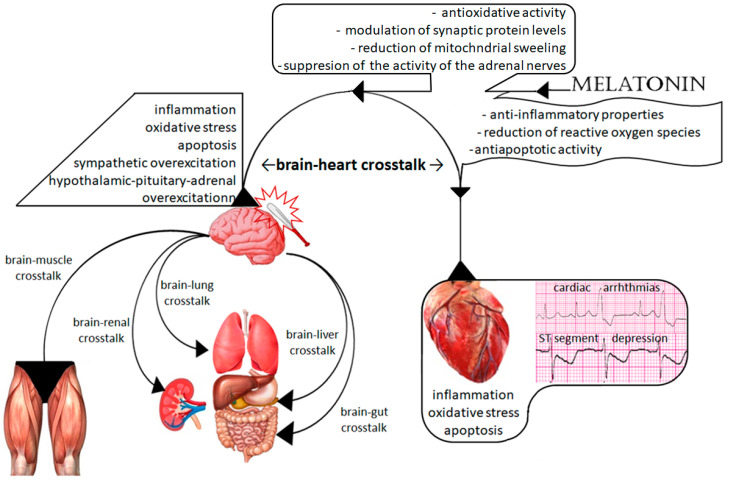
Scheme of brain–multiorgan interaction and the action of melatonin.

**Figure 2 ijms-23-07094-f002:**
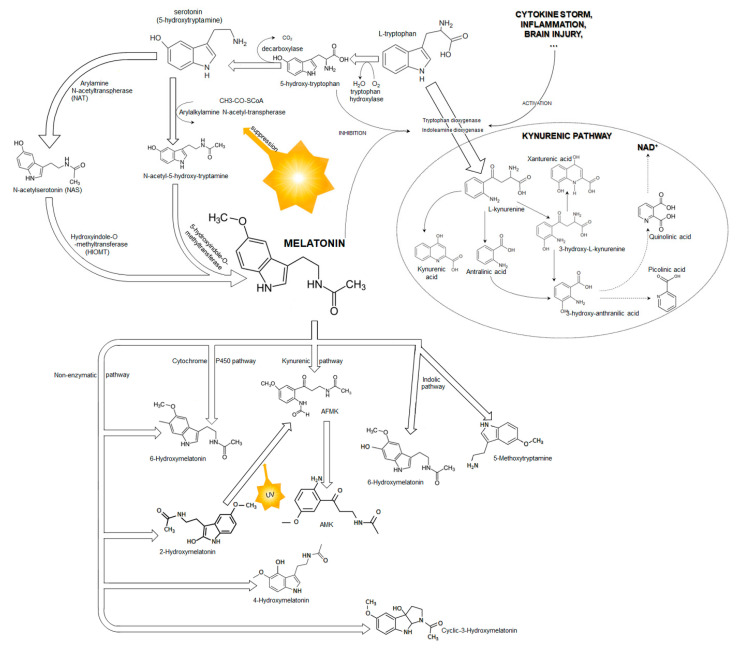
A scheme of melatonin biosynthesis from tryptophan and melatonin metabolism.

## Data Availability

Not applicable.

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
