# Peer review of "Melatonin and the Brain–Heart Crosstalk in Neurocritically Ill Patients—From Molecular Action to Clinical Practice"

_ijms, 2022, doi:10.3390/ijms23137094_

Round 1

Reviewer 1 Report

In the current review, the authors systemically discussed a sound topic i.e., melatonin and the Brain-Heart Crosstalk in neurocritical Patients – from molecular action to clinical practice. This is an important topic with significant clinical implications. In addition, this aspect has not been often discussed. Several issues should be addressed to further improve the quality of the review.

1.       In the Introduction, the authors stated that “Melatonin is synthesized from tryptophan by which is hydroxylated to 5-hydroxytryptophan and this process is stimulated by darkness” is not correct. The rate-limiting enzyme for melatonin synthesis is AANAT which is regulated by the light/dark cycle (Tan DX, Reiter RJ An evolutionary view of melatonin synthesis and metabolism related to its biological functions in plants. J Exp Bot. 2020 Aug 6;71(16):4677-4689. doi: 10.1093/jxb/eraa235). This issue should be clarified.

2.       Melatonin is synthesized in mitochondria and majority of its neuroprotective effects may occur in the level of mitochondria, especially may be mediated by the melatonin receptor located in the mitochondria. This novel and important aspect should be discussed in details to improve the quality of the review (references: Tan, D.-X. and Reiter, R.J. 2019. Mitochondria: the birth place, battle ground and the site of melatonin metabolism in cells. Melatonin Research. 2, 1 (Feb. 2019), 44-66. DOI:https://doi.org/https://doi.org/10.32794/mr11250011.

3.       In figure 2, the enzyme “N-acetyltransferase” is incorrect. N-acetyltransferase is a nonspecific enzyme of acetyltransferase. The correct enzyme is arylalkylamine N-acetyltransferase (AANAT) which is responsible for melatonin synthesis. Please correct it.

4.       In the Abstract: “inti-inflammatory” should be “anti-inflammatory”; “ani-apoptotic” should be “anti-apoptotic”.

5.       The English should be polished buy a native English speaker.

Author Response

Ms. Ref. No.:  JECG-D-16-00208

Dear Editor,

We would like to thank the reviewers for their careful and helpful review of our manuscript titled: "Melatonin and the Brain-Heart Crosstalk in neurocritically Ill Patients – from Molecular Action to Clinical Practice". Based on the comments received, we have now further improved the manuscript per the reviewers' suggestions, marking all changes and corrections in red throughout the text. Our point-by-point responses to the reviewers’ helpful comments are also shown below.

Response to Reviewers:

Reviewer #1:

Comment 1:  In the Introduction, the authors stated that “Melatonin is synthesized from tryptophan by which is hydroxylated to 5-hydroxytryptophan and this process is stimulated by darkness” is not correct. The rate-limiting enzyme for melatonin synthesis is AANAT which is regulated by the light/dark cycle (Tan DX, Reiter RJ An evolutionary view of melatonin synthesis and metabolism related to its biological functions in plants. J Exp Bot. 2020 Aug 6;71(16):4677-4689. Doi: 10.1093/jxb/eraa235). This issue should be clarified.

We thank the reviewer for this important comment, and we agree. We corrected our text and figure 2.  

Comment 2: Melatonin is synthesized in mitochondria and majority of its neuroprotective effects may occur in the level of mitochondria, especially may be mediated by the melatonin receptor located in the mitochondria. This novel and important aspect should be discussed in details to improve the quality of the review (references: Tan, D.-X. and Reiter, R.J. 2019. Mitochondria: the birth place, battle ground and the site of melatonin metabolism in cells. Melatonin Research. 2, 1 (Feb. 2019), 44-66. DOI:https://doi.org/https://doi.org/10.32794/mr11250011.

Thanks for this helpful comment. We added a small paragraph.

Comment 3.   In figure 2, the enzyme “N-acetyltransferase” is incorrect. N-acetyltransferase is a nonspecific enzyme of acetyltransferase. The correct enzyme is arylalkylamine N-acetyltransferase (AANAT) which is responsible for melatonin synthesis. Please correct it. Thank you for this comment. We corrected it.

Comment 4. In the Abstract: “inti-inflammatory” should be “anti-inflammatory”; “ani-apoptotic” should be “anti-apoptotic”. Thank You for this comment. All grammatical errors were corrected.

Reviewer 2 Report

This is an interesting review that requires revisions, see below.

Figure 2 contains errors. It is incorrect to connect melatonin into kynurenic pathway of tryptophan degradation.

Melatonin metabolism/degradation deserves separate panel showing indolic and kynuric routes of metabolism (cf Exp Dermatol 26:563–568,  2017; Cell Mol Life Sci 74(21), 3913-3925, 2017). In fact enzymatic and non-enzymatic routes should be discussed including UVB radiation (FASEB J  20, 1564-1566, 2006)

Enzymatic metabolism is very rapid (FASEB J 27, 2742–2755, 2013) which may affect phenotypic activity.

Better overview of melatonin synthesis is expected. Note that acetylation of serotonin can be performed by AANAT and alternative NAT (Eur J Biochem 270, 3335-3344, 2003; FASEB J 19, 176-194, 2005).

Role of NAS could also be mentioned (J Pineal Res, 2020 Mar;68(2):e12626. doi: 10.1111/jpi.12626).

More in depth mechanism of action could be discussed. Note, that in addition of receptor mediated (by MT1 and MT2) some actions of melatonin such as induction of anti-oxidative responses are independent from MT1 and MT2 (Sci Rep 2017;7(1):1274, 2017). 

Mention that outside of brain melatonin is produced in many peripheral organs some of them exposed to environmental stress such as skin.

Author Response

Ms. Ref. No.:  JECG-D-16-00208

Dear Editor,

We would like to thank the reviewers for their careful and helpful review of our manuscript titled: "Melatonin and the Brain-Heart Crosstalk in neurocritically Ill Patients – from Molecular Action to Clinical Practice". Based on the comments received, we have now further improved the manuscript per the reviewers' suggestions, marking all changes and corrections in red throughout the text. Our point-by-point responses to the reviewers’ helpful comments are also shown below. 

Reviewer #2:

Comment 1: Figure 2 contains errors. It is incorrect to connect melatonin into kynurenic pathway of tryptophan degradation. Thanks for this comment. We connected melatonin and kynurenic pathways process because disorders in one of them may affect each other. Melatonin is synthetized from tryptophan and massive inflammatory response with cytokine storm (particularly interferon gamma) intensified tryptophan metabolism in kynurenine pathway reducing its available for serotonin and melatonin synthesis. Additionally, several studies documented that melatonin can inhibit IDO and TDO activity and the subsequent production of L-kynurenine (Curr Opin Immunol 2021;70:7-14, FEBS J 2017;284948-966, Chem Soc Rev 1995;24:401-412, Exp Rev Mol Med 2006;8(20) doi.10.1017/S1462399406000068). Additionally, TDO is completely inhibited by melatonin. Therefore, we connected these pathways. We also noted this fact in our manuscript.

Comment 2: Melatonin metabolism/degradation deserves separate panel showing indolic and kynuric routes of metabolism (cf Exp Dermatol 26:563–568,  2017; Cell Mol Life Sci 74(21), 3913-3925, 2017). In fact enzymatic and non-enzymatic routes should be discussed including UVB radiation (FASEB J  20, 1564-1566, 2006). Enzymatic metabolism is very rapid (FASEB J 27, 2742–2755, 2013) which may affect phenotypic activity. We thank for this helpful comment. We corrected our manuscript and added some sentences describing the metabolism of melatonin. 

Comment 3: Better overview of melatonin synthesis is expected. Note that acetylation of serotonin can be performed by AANAT and alternative NAT (Eur J Biochem 270, 3335-3344, 2003; FASEB J 19, 176-194, 2005). Thanks for this comment. We corrected the part of melatonin synthesis.

Comment 5: Role of NAS could also be mentioned (J Pineal Res, 2020 Mar;68(2):e12626. Doi: 10.1111/jpi.12626). We thank the reviewer for this important comment. We added a few sentences describing the role of NAS.

Comment 6: More in-depth mechanism of action could be discussed. Note, that in addition of receptor mediated (by MT1 and MT2) some actions of melatonin such as induction of anti-oxidative responses are independent from MT1 and MT2 (Sci Rep 2017;7(1):1274, 2017). We thank for this helpful comment. We added some sentences describing others than receptors-related mechanism of melatonin activity

Comment 7: Mention that outside of brain melatonin is produced in many peripheral organs some of them exposed to environmental stress such as skin. We thank You for this very important comment. Indeed, skin plays a very important role in melatonin synthesis and metabolism. We added some sentence to our text. 

Round 2

Reviewer 2 Report

The effort of authors to improve the presentation is appreciated.

However, disappointing is lack of attention to detail, which exposes limited expertise in melatonin biology. Further marked by factual errors. The authors claim that the corrected in reply to critique but I see still errors documenting that they did not read the suggested articles, and failed to properly refer the information, or referred it incorrectly

Melatonin synthesis pathway still contains errors. For example serotonin is acetylated also by alternative enzyme (NAT), which is not corrected, see previous comment 3 plus proper citations.

Comment 2 is not addressed properly. Melatonin metabolism is described poorly, please correct this with proper referral.

Comment 5, despite authors claim in reply is addressed. I do not see information that NAS is entering circulation and has biological activity with proper referral.

Comment 6 is not addressed properly in the manuscript. Also I do not understand while so attention is given to MT3, which is not recognized by many experts as the receptor.

Importantly the sentence 76-77 is incorrect. Pineal gland has higher concentrations of melatonin than keratinocytes. The cited paper by Fisher was on immortalized cultured cells in media with serum, therefore, cannot be used as representative of melatonin content in the epidermal keratinocytes. In addition its main objective was to show melatonin metabolism, which was endogenous or UVB induced. The proper citation for melatonin content in epidermal keratinocytes is in Kim et al., Mol Cell Endocrinol 404, 1-8, 2015.

There are also statements without proper citations or with incorrect citations.

These significant errors and lack of attention to detail decrease the enthusiasm for this manuscript, which should represent state of the art knowledge in the field.

Author Response

Dear Reviewer,

We would like to thank You for Your very helpful review of our manuscript. Based on the comments we corrected our manuscript and all new changes we marked as a blue colour. Red colour is the first correction. We hope our corrections will satisfy You.

Melatonin synthesis pathway still contains errors. For example serotonin is acetylated also by alternative enzyme (NAT), which is not corrected, see previous comment 3 plus proper citations. Thank You very much for this comment. We add some sentence to our text. We also changed the Figure 2.

Comment 2 is not addressed properly. Melatonin metabolism is described poorly, please correct this with proper referral. Thank You very much for this comment. We tried to described melatonin metabolism more detail. We hope the present version is more suitable for our manuscript. 

Comment 5, despite authors claim in reply is addressed. I do not see information that NAS is entering circulation and has biological activity with proper referral. Thank You very much. We added a few sentence describing synthesis and activity of NAS.

Comment 6 is not addressed properly in the manuscript. Also I do not understand while so attention is given to MT3, which is not recognized by many experts as the receptor. We mentioned about MT3 only in one sentence. We know, that there are two receptors for melatonin. 

Importantly the sentence 76-77 is incorrect. Pineal gland has higher concentrations of melatonin than keratinocytes. The cited paper by Fisher was on immortalized cultured cells in media with serum, therefore, cannot be used as representative of melatonin content in the epidermal keratinocytes. In addition its main objective was to show melatonin metabolism, which was endogenous or UVB induced. The proper citation for melatonin content in epidermal keratinocytes is in Kim et al., Mol Cell Endocrinol 404, 1-8, 2015. Thank You very much for this comment. We. We deleted this sentences and changed this paragraph generally.

There are also statements without proper citations or with incorrect citations. Thank You very much. We check and correct all citations carefully.

On behalf of all co-authors

Wojciech Dabrowski

Round 3

Reviewer 2 Report

The authors adequately revised the manuscript